# Aggression in Women with Schizophrenia Is Associated with Lower HDL Cholesterol Levels

**DOI:** 10.3390/ijms231911858

**Published:** 2022-10-06

**Authors:** Dora Herceg, Ninoslav Mimica, Miroslav Herceg, Krešimir Puljić

**Affiliations:** 1School of Medicine, University of Zagreb, Šalata 3, HR-10000 Zagreb, Croatia; 2University Psychiatric Hospital Vrapče, Bolnička Cesta 32, HR-10090 Zagreb, Croatia

**Keywords:** aggression, female patients, lipids, schizophrenia

## Abstract

This study assessed the association between serum lipid levels and aggression in female patients with schizophrenia. The study included female patients with schizophrenia (N = 120). The participants were subdivided into two groups (aggressive and nonaggressive), with 60 participants in each group. Serum lipids—cholesterol, triglycerides, high-density lipoproteins (HDL cholesterol), and low-density lipoproteins (LDL cholesterol)—were determined. The clinical part of the study included an evaluation using psychiatric scales: the positive and negative syndrome scale (PANSS), the aggression subscale of the PANSS scale (PANSS-AG), and the overt aggression scale (OAS). Significant differences were only observed in HDL cholesterol levels, where aggressive subjects had significantly lower values of HDL cholesterol (t = 2.540; *p* = 0.012), and the representation of subjects with low cholesterol values was almost three-times higher in the group of subjects with aggression (χ^2^ = 7.007; *p* = 0.008) compared to the nonaggressive group. The nominally significant predictor for HDL cholesterol in nonaggressive and aggressive participants was the total value of the PANSS scores. In subjects with aggression, suicidality was not significantly associated with HDL cholesterol levels. Our findings suggest that lower HDL cholesterol is significantly associated with aggression in women with schizophrenia.

## 1. Introduction

Patients with schizophrenia have a higher tendency to be aggressive than the general population. The causes that can lead to aggression in schizophrenia are also more numerous than for heterogeneous persons. The aggressive behavior of patients with schizophrenia has serious clinical and social consequences and represents a major burden in the care and treatment of such patients. The simplest division of aggression is into auto-aggressiveness (self-harm, suicidality) and hetero-aggressiveness (aggression directed towards others) [1]. This research will present the complex issue of the relationship between lipids and aggressive behavior in schizophrenia. In a previous study, we found a significant association between high aggression and high prolactin levels (PRL) in female patients with schizophrenia [2].

Lipids participate in various physiological functions of the organism. They are essential components of cell membranes, a storehouse of metabolic energy, and very important signaling molecules in cellular metabolism. The major particles of serum lipoproteins are chylomicrons, very low-density lipoproteins (VLDL), low-density lipoproteins (LDL), medium-density lipoproteins (IDL, intermediate-density lipoproteins), and high-density lipoproteins (HDL) [3,4].

Cholesterol, together with sphingolipids and gangliosides, is combined into hydrophobic micro-domains of the membrane, popularly called lipid rafts. They are responsible for several cellular processes—for example, coordination and signaling, endocytosis, and communication with the cytoskeleton. As cholesterol is an essential component of lipid rafts, any change in cholesterol content can affect the aforementioned processes, thus altering synaptic transmission and nerve plasticity. Therefore, a lack or excess of cholesterol in the brain can have very serious consequences. Changes in membrane lipid rafts could explain abnormalities in serotonin and dopamine neurotransmission, which are associated with cholesterol depletion. Decreased levels of the serotonin metabolite 5-hydroxyindoleacetic acid (5-HIAA) are associated with depression, suicidal behavior, and aggression, which altogether lead to brain disorders [5,6]. Cholesterol is likely to act as a moderator of serotonin function [7].

Changes in serum lipid levels were found in chronic patients on antipsychotic therapy as well as in patients who had experienced their first episode of psychosis and were not yet receiving any medication [8,9,10]. Of course, there are also studies from which contradictory or mixed results emerge on the association of changes in peripheral lipid levels and the early course of schizophrenia. Wysokinski et al. found that elevated values of total cholesterol, LDL cholesterol, triglycerides, and glucose and decreased values of HDL cholesterol are present in schizophrenia and bipolar and unipolar depressive disorder [11]. Patients with schizophrenia had higher serum triglyceride levels and lower serum HDL cholesterol levels than healthy controls [11]. It is hypothesized that these changes in lipid profile affect clinical characteristics, disease symptom intensity, and treatment prognosis [12].

A recent review [13] included 23 studies that evaluated the association between lipid levels and aggression, violence, or suicidal behavior. Suicidal behavior and plasma lipid levels were assessed in 20 studies, and some studies reported low cholesterol levels in suicidal subjects, while others did not confirm these results. Aggression towards others was assessed in nine studies that found an association between plasma lipid levels and aggression, but the results were very divergent [13]. The authors concluded that the majority of studies reported an association between low cholesterol and violence towards others but stressed that not only cholesterol but other lipid parameters, such as HDL cholesterol, LDL cholesterol, and triglycerides, should also be investigated; they also stated that there is mixed evidence for an association between low cholesterol and suicidality in schizophrenia. In addition, they concluded that only a few studies evaluated sex differences and they yielded mixed evidence [13].

In 1990, Muldoon et al. published a paper in which they concluded that lowering serum cholesterol in middle-aged patients with medication, diet, or a combination of drugs and diet led to a reduction in coronary heart disease but also to increased mortality from suicide and violent behavior [14]. In 1992, Engelberg was the first to hypothesize an association between plasma levels of total cholesterol, the content of cholesterol in brain cell membranes, the serotonin system, and behavioral predispositions. One of the roles of serotonin in the central nervous system is the suppression of harmful impulsive behaviors [15]. When the synaptosomal membrane content of cholesterol in the brain is increased, an increase in the number of serotonin receptors is also expressed. This is why lowering serum cholesterol can reduce the proportion of cholesterol in brain cell membranes as well as lipid micro-viscosity, and it can also reduce exposure to protein serotonin receptors on the membrane surface [16]. All of this results in poorer intake of serotonin from the circulation, which means that less serotonin enters the brain cells. This contributes to a decrease in brain serotonin function, while low cholesterol reduces serotoninergic receptor activity, which leads to poor suppression of aggressive behavior [15], leading to increased aggression, impulsivity, and suicidal behavior [13].

## 2. Results

It can be seen from Table 1, as reported previously [2], that the age of the subjects and the duration of the disease were approximately equal in the groups of aggressive and nonaggressive subjects. Nominally significant differences were observed in the length of hospital stay, which was longer in aggressive subjects (t = −2.287; *p* = 0.024, Student’s *t*-test), while the received dose of antipsychotics expressed in chlorpromazine equivalents was, on average, three-fold higher in aggressive subjects (t = −6.533; *p* = 1.7 × 10^−9^, Student’s *t*-test). The number of women who smoked did not differ significantly (χ^2^ = 0.154; *p* = 0.422, Chi-square test) between the two groups of patients, but the number of cigarettes smoked per day was slightly higher in the group of aggressive patients (t = −2.380; *p* = 0.020, Student’s *t*-test).

The total values of scores on the PANSS scale (t = 4.905; *p* < 0.001), as well as on the scales of positive (t = 4.648; *p* < 0.001), negative (t = 3.225; *p* = 0.002), and general psychopathology (t = 3.629; *p* < 0.001) symptoms, were significantly higher in aggressive subjects (Table 1). As nonaggressive subjects had a minimum number of scores on the extended PANSS scale for aggression (PANSS-AG) and on the OAS scale, these values were not included in the statistical analysis. It was observed that nonaggressive subjects did not show symptoms of suicidal behavior, in contrast to aggressive subjects, among whom the proportion of suicidal subjects was 30% (Table 1).

Table 2 shows the values of total, HDL, and LDL cholesterol, as well as the triglyceride concentrations, in the blood of subjects with schizophrenia with and without aggression. In addition, the subjects were divided into those who had levels of these metabolic parameters above, below, or within the reference intervals, so the reference value for cholesterol was 5 mmol/L, for triglycerides was 1.7 mmol/L, for HDL cholesterol was 1.2 mmol/L, and for LDL cholesterol was 3 mmol/L. The prevalence of subjects with elevated cholesterol, elevated triglycerides, and elevated LDL cholesterol was not significantly different between these two groups of subjects (Table 2). Significant differences were observed only in the values of HDL cholesterol, where aggressive subjects had significantly lower values of HDL cholesterol (t = 2.540; *p* = 0.012) (Table 2, Figure 1), and the representation of subjects with low cholesterol values was almost three-times higher in the group of subjects with aggression (χ^2^ = 7.007; *p* = 0.008) (Table 2).

HDL cholesterol levels in subjects with and without aggression were tested for the influence of age, received dose of therapy, smoking, severity of symptoms of schizophrenia, aggression, and the presence of suicidal behavior by multiple linear regression (Table 3). The only predictor in both groups of respondents (nonaggressive: *p* = 0.024 and aggressive: *p* = 0.022) was the total value of scores on the PANSS scale, but, due to the correction for multiple testing, its effect remained marginally significant. Other tested variables did not show a significant effect on HDL cholesterol levels in any group of subjects in this model.

Additionally, the presence of aggression in subjects with schizophrenia was tested for the effect of confounding variables (age, PANSS total scores, and smoking), as well as lipid levels, with logistic regression (Table 4). Higher PANSS scores were significant predictors (*p* < 0.001) of aggression, while lower levels of HDL cholesterol, due to correction for multiple testing, contributed to the model with marginal significance (*p* = 0.047).

Significant correlations between HDL cholesterol levels and aggression severity were not observed in relation to the total scores on either the extended PANSS aggression scale (r = −0.005; *p* = 0.704) or on the OAS scale (r = 0.001; *p* = 0.999). Moreover, the total scores on the OAS (t = 0.532; *p* = 0.633) and PANSS-AG (t = −0.309; *p* = 0.758) scales were not significantly different between subjects with low HDL cholesterol levels and those with HDL cholesterol in terms of reference values.

Suicidality in subjects with aggression was not significantly associated with HDL cholesterol levels; namely, both suicidal and non-suicidal subjects had approximately equal levels of HDL cholesterol, and subjects with low cholesterol were equally represented in both groups (Table 5).

To control for the possible influence of the confounding variables on suicidal behavior in aggressive subjects, the effect of age, PANSS total scores, smoking, and lipid levels on suicidal behavior was examined with logistic regression (Table 6). None of the tested variables contributed significantly to the model.

## 3. Discussion

Regarding demographic and clinical data, the results show that significant differences were observed in the duration of hospitalization, which was significantly longer in aggressive subjects, and in the received dose of antipsychotics expressed in chlorpromazine equivalents, with aggressive subjects receiving, on average, approximately three-times higher doses of antipsychotics. The number of cigarettes smoked per day was significantly higher in the group of aggressive patients. Moreover, the total scores on the PANSS scale, as well as on the scales of positive, negative, and general symptoms, were significantly higher in aggressive subjects. Based on these results, aggressive patients had a more severe clinical picture, needed a higher dose of antipsychotics in treatment, and had a longer duration of treatment. Logistic regression revealed that the total PANSS scores were significant predictors of aggression. In line with our results, aggression was associated with psychotic disorders, and higher scores on the PANSS were related to aggression [17]. Therefore, psychotic patients, especially those with schizophrenia, had an increased risk of becoming aggressive on the psychiatric ward [17]. As for the higher number of cigarettes smoked per day in aggressive patients, this may be associated with self-medication, as nicotine has a calming effect and alleviates the side effects of antipsychotics. Regarding the influence of age and gender, the subjects were exclusively female patients and there was no difference in age between the two groups (aggressive and nonaggressive patients).

In contrast to our data, age was a significant factor in a previous study that evaluated the association of lipids and suicidality, a meta-analysis consisting of 65 epidemiological studies [18]. It showed that the association between lower serum lipid levels and suicidality was stronger in individuals younger than 40 years compared to older participants [18].

Gender also seems to play an important role since suicidal women, but not suicidal men, had lower HDL cholesterol compared to control subjects in a prior study [18]. In contrast to these results, Svensson et al. reported an association between elevated serum cholesterol levels and suicidality in Japanese women [19]. Violent behavior is more common in men than in women [7]. Men appear to be more sensitive than women to low cholesterol, given the association of low cholesterol and aggression found mostly in men [7]. Vevera et al. conducted a retrospective study that included only the female population. The results showed that women who attempted violent suicide had significantly lower total cholesterol levels compared with women who attempted nonviolent suicide [20]. As recently reviewed [17], there are inconsistent data on the role of gender in aggression in the psychiatric ward: some studies (*n* = 12) confirmed male gender as a risk factor for aggression, others found no relation (*n* = 22), and some studies (*n* = 3) reported that female gender is a risk factor for aggression. Therefore, it is still unclear whether male or female gender is a risk factor for aggression [13]. Since these data are not uniform, in the present study, we included only the female gender in order to exclude the possible influence of gender on aggression.

In our study, significantly lower values of LDL cholesterol were found in the group of female patients with aggressive behavior compared to nonaggressive patients with schizophrenia. Other lipid fractions—cholesterol, triglycerides, or LDL cholesterol—were not significantly different between the two groups. In addition, the representation of subjects with low cholesterol levels was almost three-times higher in the group of subjects with aggression than in the group without aggression. This is considered to be the most important result of this study. Our results are not in line with the findings obtained in large groups of patients with schizophrenia and bipolar spectrum disorders, where no significant associations between HDL cholesterol levels and aggression or impulsivity were detected [21]. The level of HDL cholesterol in subjects with and without aggression was tested for the influence of age, primary dose of therapy, smoking, severity of symptoms of schizophrenia and aggression, and the presence of suicidal behavior, by multiple linear regression. The only nominally significant predictor in both groups of respondents (nonaggressive and aggressive) was the total number of scores on the PANSS scale. Other tested variables did not show a significant effect on HDL cholesterol levels in any group of subjects in this model. The lack of significant changes in cholesterol, triglycerides, and LDL levels between aggressive and nonaggressive patients is in line with the absence of significant associations between lipid levels and aggression or impulsivity in psychotic patients [21].

Suicidality in subjects with aggression was not significantly associated with HDL cholesterol values, or with other confounders (age, severity of symptoms of schizophrenia, smoking, and lipid levels). Namely, both suicidal and non-suicidal female patients had approximately equal levels of HDL cholesterol, and subjects with low cholesterol levels were equally represented in both groups. In contrast, in a study by Wu et al. [18], suicidal women, but not men, had significantly lower HDL cholesterol levels compared with healthy controls. These differences might be explained by the smaller number of female patients included in our study, compared to the cited meta-analysis [18]. Other studies reported contradictory findings, i.e., male suicidal vs. non-suicidal patients who had experienced their first episode of psychosis had lower cholesterol levels [22], or men were more prone to aggression if they had low cholesterol levels [7]. A meta-analysis found that suicidal men had significantly lower triglyceride values than healthy male controls, but statistical significance did not show the same trend in women [18]. Many studies that have examined the association between cholesterol and impulsivity, or aggression, generally involved only men, or the association itself was found only in men [23]. Ainiyet and Rybakowski published a study in Poland that studied a possible link between suicidal behavior and lipid levels in people with schizophrenia who were hospitalized in a psychiatric hospital due to an acute exacerbation of their mental condition [24]. A significant association was found between suicidal thoughts and suicide attempts with low levels of total and LDL cholesterol, triglycerides, and total lipids [24]. According to our results, no association was observed between HDL cholesterol or other plasma lipids and suicidality.

Our study did not show a significant association between triglyceride concentration in aggressive subjects and the severity of aggression as assessed by the OAS and PANSS-AG aggression scales. The results suggest that triglycerides, as a subset of lipids, do not display as important a role or association with aggression as cholesterol does.

The lipid profiles of hospitalized psychiatric patients with various forms of psychosis showed that there was a link between the measured low values of total cholesterol and violence, i.e., aggression [21,22,23,24]. The results of the measured values of triglycerides, LDL cholesterol, and HDL cholesterol were not so unambiguously associated with aggression [21,22,23,24,25,26,27]. Most of the results suggest that reduced levels of total cholesterol, LDL cholesterol, and triglycerides may be associated with more pronounced aggression through decreased central serotonin activity, by modulating various serotonin receptors [16,18].

Altered lipid homeostasis in the brain may particularly affect the serotonin system through the effects of cholesterol on lipid rafts on the cell membrane [16,18]. A significantly lower level of HDL cholesterol could be a potential biological marker of aggressive behavior in women with schizophrenia. Many studies to date have confirmed that individuals with low cholesterol levels are more prone to aggressive, violent, and suicidal behavior than those with cholesterol levels within the reference range [7,13].

Our findings of lower HDL cholesterol in aggression are supported by recent findings [28]. A study, using untargeted metabolomics in schizophrenia, revealed that 19 metabolites, associated with amino acid metabolism (vanillyl mandelic acid, malic acid, 4-hydroxyphenylpyruvic acid, 4-hydroxy-L-proline, L-methionine, ratio of L-asparagine/L-aspartic acid), associated with lipid metabolism (malonic acid, glycerol, glycerol 3-phosphate, glyceraldehyde), associated with pentose phosphate pathway (D-ribose), associated with purine metabolism (uric acid), and associated with pyrimidine metabolism (3-aminoisobutanoic acid), and metabolites such as L-sorbose, 3-aminosalicylic acid, glutaric acid, 4-hydroxyproline, ribitol, and ribonolactone, could differentiate violent vs. nonviolent patients with schizophrenia. Therefore, the authors discussed their significant finding of a dysregulated lipid metabolism in violent schizophrenic patients, confirming previous findings of lower fatty acid levels in patients displaying aggression or violence, and a significant correlation between low cholesterol levels and aggression or violence [28]. In addition, this study showed that metabolic pathways including phenylalanine, tyrosine, and tryptophan biosynthesis differed significantly between violent and nonviolent patients with schizophrenia. Dysregulated lipids and amino acids were related to a specific metabolic phenotype of violent schizophrenia, and these findings are in line with the previous results showing that reduced lipids (cholesterol) are associated with lower peripheral serotonin concentrations [22], resulting in aggressive or suicidal behavior.

The limitations of the study are that although most of the confounding factors were controlled, diet and exercise were not controlled, despite the fact that they might affect serum lipid levels. Since all subjects were inpatients from the same ward of the same hospital, they received the same diet and had a similar daily routine of walking, and therefore we excluded variations due to diet and exercise, or the different hospital settings [17]. Due to the multiple testing, the *p*-value was adjusted for multiple analyses of different lipid levels’ effects on aggression and suicidality. A limitation is that we did not include a control group of healthy subjects. As this was not a longitudinal study, causality could not be established. Therefore, a cautious interpretation of these results is needed, also due to the limited sample size.

For further studies, confirmation in larger and more diverse groups, such as participants with other psychotic disorders, and the inclusion of male participants and/or healthy control groups is recommended.

In conclusion, this study aimed to evaluate the association between serum lipid levels (cholesterol, triglycerides, HDL cholesterol, and LDL cholesterol) and aggression or suicidality in female patients with schizophrenia. Aggressive patients with schizophrenia had significantly lower HDL cholesterol levels, and the representation of subjects with low cholesterol values was almost three-times higher, compared to nonaggressive patients. In patients with aggression, suicidality was not significantly associated with HDL cholesterol levels. Aggression and suicidality were controlled for the possible influence of confounders (age, PANSS total scores, smoking, and lipid levels).

Our findings suggest that lower HDL cholesterol is significantly associated with aggression towards others in female patients with schizophrenia. A major contribution of these results is that serum HDL cholesterol might be used as a biomarker of aggression in female schizophrenic patients. These findings may contribute to the prevention of aggression in schizophrenia patients by using HDL cholesterol as a predictive factor.

The strengths of this study are in the inclusion of an ethnically homogeneous group of female patients with schizophrenia from the same hospital, diagnosed and evaluated by the same psychiatrists; in the well-characterized sample; in the control of potential confounding factors such as age, severity of symptoms of schizophrenia (PANSS), smoking, and lipid levels, and the correction for multiple comparisons.

## 4. Materials and Methods

### 4.1. Participants

The study included female patients with schizophrenia (N = 120) hospitalized at the University Psychiatric Hospital Vrapče from 1 March 2017–1 March 2019, as reported previously [2]. The diagnosis was made according to the criteria of the DSM-5. The inclusion criteria were as follows: a diagnosis of schizophrenia, age between 18 and 45 years old, female gender, and the absence of any other psychiatric and neurological disorder. Exclusion criteria were as follows: subjects with previous hormonal disorders, other somatic diseases, or subjects receiving hypolipidemic therapy (statins). The participants were divided into two groups, with 60 participants in each group. In the first group, the indication for hospitalization was aggression, regardless of whether the aggression was directed towards others or towards oneself (so it manifested itself as suicidal behavior). In the second group, which was also a control group of people with schizophrenia, the indication for hospitalization was worsening of schizophrenia, but without aggression. The clinical part of the study included a psychiatric interview and evaluation using psychiatric scales within 48 h of hospital arrival. As described previously [2], the severity of the clinical picture of schizophrenia was evaluated by the positive and negative syndrome scale (PANSS) and the PANSS subscales (positive, negative, and general psychopathological subscales) [29]. The aggressiveness of the subjects was graded using the overt aggression scale (OAS), which divides aggression into 4 subgroups; the OAS scale determines the intensity of each of the possible forms of aggression with 1 to 6 points (0 = none, 6 = severe personal injuries) [30]. We also used the aggression subscale of the PANSS scale (AG PANSS), which includes supplementary components for the risk aggression profile (such as anger, resentment, difficulty in delaying gratification, and affective lability) [29]. Insight into the medical history of each participant provided information on current and previous treatments, number of hospitalizations, therapy and duration of illness, previous suicide attempts, and hetero-aggressiveness. In addition, the influence of confounders (age, smoking, suicidal behavior) was monitored and controlled in the study. The purpose and goal of the study was explained to all participants and they provided informed consent. The Ethics Committee of the University Psychiatric Hospital Vrapče gave permission to conduct this study.

### 4.2. Determination of Serum Lipid Levels (Cholesterol, Triglycerides, HDL Cholesterol, LDL Cholesterol)

Lipid levels were measured within 48 h of admission to the hospital. To determine the levels of total cholesterol, triglycerides, HDL cholesterol, and LDL cholesterol, blood was drawn from the cubital vein of all subjects in the period from 07:00 to 09:00 am, after a night of fasting. All lipids were determined on a Cobas Integra 400 plus biochemical analyzer (Roche Diagnostics, Mannheim, Germany), using original Roche Diagnostics reagents. Total cholesterol was determined by a photometric method, the CHOD-PAP procedure. Triglycerides were also determined by a photometric method, the GPO-PAP procedure. HDL cholesterol was determined by a homogeneous enzymatic method with polyethylene glycol and dextran sulfate. LDL cholesterol was determined computationally using the Friedewald equation, if the triglyceride concentration was up to 4 mmol/L. The concentrations of the tested lipids were read from the corresponding calibration curves.

### 4.3. Statistical Analysis

Statistical data processing was performed in SigmaStat 3.5 (Jandel Scientific Corp., San Jose, CA, USA). The normality of the distribution was examined for each of the tested variables by the Kolmogorov–Smirnov test. Since most of the data were normally distributed, all data were presented as mean values and corresponding standard deviations (SD), and parametric statistical tests were used for analysis. Multiple linear regression was used to examine the effects of multiple independent variables on the dependent variable (cholesterol, triglyceride, HDL cholesterol, and LDL cholesterol levels), while logistic regression was used to predict the aggression and suicidal behavior outcome depending on the confounding variables. Differences in tested variables between two independent groups were evaluated with Student’s *t*-test. The Pearson correlation test examined the association of two continuous numerical variables (dependence of the number of points on psychometric scales with metabolic parameters), while the χ^2^ test was used to test the distribution of categorical variables within 2 or more groups. The significance level was set at *p* < 0.0125, due to testing for four lipid parameters, with all tests being bidirectional.

## 5. Conclusions

In this study, it was found that lower HDL cholesterol is significantly associated with aggression in female patients with schizophrenia.

## Figures and Tables

**Figure 1 ijms-23-11858-f001:**
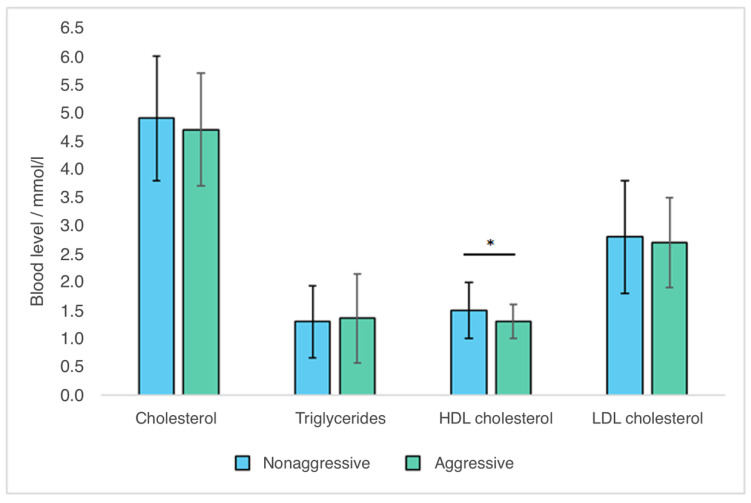
Relationship between the values of metabolic parameters and aggression in subjects with schizophrenia. Data are presented as mean ± standard deviation (SD); * *p* = 0.012 vs. HDL cholesterol concentration in nonaggressive subjects, Student’s *t*-test.

**Table 1 ijms-23-11858-t001:** Demographic and psychometric data in female subjects with schizophrenia divided into nonaggressive and aggressive subjects.

Subjects with Schizophrenia				
	Nonaggressive (*N* = 60)		Aggressive (*N* = 60)	Statistics
Age (years)		37 ± 8	35 ± 7	t = 1.383; *p* = 0.169
Length of illness (years)		9 ± 6	8 ± 6	t = 0.183; *p* = 0.855
Length of hospitalization (days)		24 ± 8	28 ± 10	t = 2.287; *p* = 0.024
Antipsychotic dose * (mg/day)		180.9 ± 115.5	343.8 ± 154.7	t = 6.533; ***p* < 0.001**
Smoking (N; %)	Yes	40 (66.7%)	42 (70.0%)	χ^2^ = 0.154; *p* = 0.422
	No	20 (33.3%)	18 (30.0%)	
Number of cigarettes (/day)		11 ± 9	16 ± 11	t = 2.380; *p* = 0.020
PANSS total scores		94 ± 9	103 ± 11	t = 4.905; ***p* < 0.001**
PANSS subscale scores for aggression		-	10 ± 5	-
OAS scores		-	12 ± 6	-
Suicidal behavior (N; %)	Yes	0 (0.0%)	18 (30.0%)	χ^2^ = 27.176; ***p* < 0.001**
	No	60 (100.0%)	42 (70.0%)	

* Doses of antipsychotics: chlorpromazine equivalents; OAS: overt aggression scale; PANSS: positive and negative syndrome scale. Data are presented as mean ± standard deviation (SD) or as number of subjects (frequency). Bold *p*-values indicate statistically significant results at the *p* < 0.0125 level.

**Table 2 ijms-23-11858-t002:** Lipid concentrations in subjects with schizophrenia depending on the presence of aggressive behavior.

Female Subjects with Schizophrenia			
	Nonaggressive (*N* = 60)	Aggressive (*N* = 60)	Statistics
Cholesterol/mmol/L	4.9 ± 1.1	4.7 ± 1.0	t = 1.320; *p* = 0.190 *
Cholesterol ≤ 5 mmol/L (N; %)	33 (55.0%)	37 (61.7%)	χ^2^ = 0.549 *p* = 0.459 **
Cholesterol > 5 mmol/L (N; %)	27 (45.0%)	23 (38.3%)	
Triglycerides/mmol/L	1.30 ± 0.64	1.36 ± 0.79	t = −0.450; *p* = 0.653 *
Triglycerides ≤ 1.7 mmol/L (N; %)	48 (80.0%)	47 (78.3%)	χ^2^ = 0.051 *p* = 0.822 **
Triglycerides > 1.7 mmol/L (N; %)	12 (20.0%)	19 (31.7%)	
HDL cholesterol/mmol/L	1.5 ± 0.5	1.3 ± 0.3	t = 2.540; ***p* = 0.012 ***
HDL cholesterol ≥ 1.2 mmol/L (N; %)	53 (38.3%)	41 (68.3%)	χ^2^ = 7.007 ***p* = 0.008 ****
HDL cholesterol < 1.2 mmol/L (N; %)	7 (11.7%)	19 (31.7)	
LDL cholesterol/mmol/L	2.8 ± 1.0	2.7 ± 0.8	t = 0.683; *p* = 0.496 *
LDL cholesterol ≤ 3 mmol/L (N; %)	38 (63.3%)	36 (60.0%)	χ^2^ = 0.141; *p* = 0.707 **
LDL cholesterol > 3 mmol/L (N; %)	22 (36.7%)	24 (40.0%)	

* Student’s *t*-test, ** Chi-square test; HDL: high density lipoprotein; LDL: low density lipoprotein; Mean (±standard deviation); Bold *p*-values indicate statistically significant results at the *p* < 0.0125 level.

**Table 3 ijms-23-11858-t003:** HDL cholesterol concentration depending on age, smoking, dose of antipsychotics, severity of symptoms of schizophrenia (PANSS), aggression (PANSS-AG, OAS), and suicidal behavior in nonaggressive and aggressive subjects with schizophrenia.

Female Subjects with Schizophrenia		
	Nonaggressive	Aggressive
Age	β = 0.081; *p*= 0.519	β = 0.255; *p* = 0.057
Antipsychotic dose *	β = 0.097; *p* = 0.461	β = −0.060; *p* = 0.718
Smoking	β = 0.208; *p* = 0.095	β = 0.135; *p* = 0.333
PANSS	β = 0.298; *p* = 0.024	β = 0.329; *p* = 0.022
PANSS-AG	-	β = −0.128; *p* = 0.455
OAS	-	β = 0.001; *p* = 0.993
Suicidality	-	β = 0.019; *p* = 0.893
Model	** R^2^ = 0.124; F = 3.079; *p* = 0.023	** R^2^ = 0.076; F = 1.694; *p* = 0.139

* Dose of antipsychotics: chlorpromazine equivalents. ** R^2^: corrected R^2^; PANSS: positive and negative syndrome scale; PANSS-AG: extended scale for aggression; OAS: overt aggression scale. Due to the correction for multiple testing, statistically significant results were determined at the *p* < 0.0125 level.

**Table 4 ijms-23-11858-t004:** Aggression in patients with schizophrenia depending on age, severity of symptoms of schizophrenia (PANSS), smoking, and lipid levels.

Aggression (YES/NO)	
Independent variable	Odds ratio (95% Confidence Interval)
Age	0.970 (0.911–1.032); *p* = 0.338
PANSS	1.224 (1.108–1.353); ***p* < 0.001**
Smoking	0.787 (0.321–1.930); *p* = 0.604
Cholesterol	1.301 (0.182–9.314); *p* = 0.794
Triglycerides	0.743 (0.249–2.217); *p* = 0.594
HDL cholesterol	0.115 (0.014–0.968); *p* = 0.047
LDL cholesterol	0.723 (0.098–5.322); *p* = 0.723
Model	* PAC = 74.2; ** R^2^ = 0.291; x^2^ = 28.943; ***p* < 0.001**

* PAC: percentage accuracy in classification; ** R^2^: Nagelkerke R^2^; PANSS: positive and negative syndrome scale. Bold *p*-values indicate statistically significant results at the *p* < 0.0125 level.

**Table 5 ijms-23-11858-t005:** HDL cholesterol concentration in aggressive subjects depending on suicidal behavior.

Aggressive Subjects Divided with Respect to Suicidal Behavior			
	Non-Suicidal (*N* = 42)	Suicidal (*N* = 18)	Statistics
HDL cholesterol (mmol/L)	1.340 ± 0.346	1.322 ± 0.364	t = 0.184; *p* = 0.854 *
HDL cholesterol ≥ 1.2 mmol/L (N; %)	29 (69.0%)	12 (66.7%)	χ^2^ = 0.033; *p* = 0.856 **
HDL cholesterol < 1.2 mmol/L (N; %)	13 (31.0%)	6 (33.3%)	

* Student’s *t*-test; ** Chi-square test; Data are presented as mean ± standard deviation (SD) or as number of subjects (frequency).

**Table 6 ijms-23-11858-t006:** Suicidal behavior in aggressive patients with schizophrenia depending on age, severity of symptoms of schizophrenia (PANSS), smoking, and lipid levels.

Suicidal Behavior (YES/NO)	
Independent variable	Odds ratio (95% Confidence Interval)
Age	0.963 (0.882–1.051); *p* = 0.401
PANSS	1.067 (0.937–1.214); *p* = 0.330
Smoking	2.367 (0.647–8.664); *p* = 0.193
Cholesterol	0.040 (0.000–5.439); *p* = 0.199
Triglycerides	3.922 (0.429–35.904); *p* = 0.226
HDL cholesterol	10.311 (0.116–919.05); *p* = 0.308
LDL cholesterol	23.409 (0.152–3604.981); *p* = 0.163
Model	* PAC = 70.0; ** R^2^ = 0.116; x^2^ = 5.137; *p* = 0.643

* PAC: percentage accuracy in classification; ** R^2^: Nagelkerke R^2^; PANSS: positive and negative syndrome scale.

## Data Availability

Not applicable.

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
