# Peer review of "Aggression in Women with Schizophrenia Is Associated with Lower HDL Cholesterol Levels"

_ijms, 2022, doi:10.3390/ijms231911858_

Round 1
Reviewer 1 Report
Expand the discussion about biochemistry of findings . See:
Reviewer 2 Report
The authors present an interesting study on the association of lipid levels with risk of suicide / aggression. Overall the paper is clearly presented. Here are my concerns or suggestions for consideration
1 1) For introduction –
I believe the authors may also mention other studies on lipid levels and suicide/aggression, and whether these results are consistent. The authors can then provide a motivation for their study in view of limitations/inconsistencies of previous works.
In particular, authors can explain why this study focuses on females and previous relevant studies on females.
2)I think it’s better to move the methods part before Results.
3) For the measurement of lipid levels, when were they measured? eg 2 days within admission, or as part of routine care of patients before admission?
4)One concern is known or unknown confounders between the high/low lipid groups. While a multiple regression was done using lipid as dependent variable, I would suggest an additional analysis, ie using suicide/aggression as outcome in a logistic regression, and lipids + other confounders as predictors.
5) Multiple testing may cause false positives, this can be at least mentioned as a limitation, or methods like FDR can be used to control for multiple testing.
6) I suggest a detailed discussion of limitations, eg temporal sequence of lipid change and suicide/aggression not determined
Causality cannot be established
confounding factors?
limited sample size
etc
Reviewer 3 Report
- A brief summary (one short paragraph) outlining the aim of the paper, its main contributions and strengths.
Article: does not look at some novel measurement, missing control group, why non-aggressive only, why not healthy subjects without any psychiatric diseases
Review: completeness of the review topic is covered, the gap in knowledge is identified- Specific comments - line74-75:
- This contributes to a decrease in brain...., why it is like that?
- line 250- how you controlled the confounders?

Round 2
Reviewer 2 Report
The authors have revised the manuscripts quite thoroughly and my comments are adequately addressed. The text may benefit from further edits to make it more fluent.
Author Response
The authors have revised the manuscripts quite thoroughly and my comments are adequately addressed.
Point 1: The text may benefit from further edits to make it more fluent.
Response 1: We have accepted this comment and decided to use one of the editing services listed at https://www.mdpi.com/authors/english to improve English writing and fluency.